# Low-dye taping may enhance physical performance and muscle activation in basketball players with overpronated feet

**Indy M. K. Ho**[1,2]*, **Anthony Weldon**[1], **Natalia C. Y. Yeung**[1], **Jim T. C. Luk**[1]

**1** Department of Sports and Recreation, Technological and Higher Education Institute of Hong Kong (THEi), Chai Wan, Hong Kong, **2** The Asian Academy for Sports and Fitness Professionals, Quarry Bay, Hong Kong

* indymankit@hotmail.com

**Data Availability Statement:** All relevant data are within the paper and its Supporting information files.

## Abstract

### Background

Low-dye taping (LTD) is widely used by athletes and medical practitioners but the research regarding its impacts on athletic performance is lacking. This study investigated the effects of using low-dye taping on plyometric performance and muscle activities in recreational basketball players with overpronated feet.

### Methods

Twelve collegiate males with at least three years basketball training experience and navicular drop (ND) value $\geq$10 mm performed the navicular drop, drop jump and countermovement jump tests. Surface electromyography of selected lower limb muscles were observed during bilateral free squat. All tests in non-taped (NT) and taped (TAP) conditions were counterbalanced using repeated crossover study design. Paired t-test with an alpha level of 0.05 and non-clinical magnitude-based decision (MBD) with standardized effects were used to analyze data.

### Results

Contact time and reactive strength index (RSI) in the TAP condition were significantly shorter (p = 0.041) and higher (p<0.01) than the NT condition respectively. No significant difference in CMJ performance between NT and TAP was observed. MBD demonstrated clear effects on both ND (standardized effect: -1.54±0.24), flight time (standardized effect: 0.24±0.30), contact time (standardized effect: -0.27±0.21), RSI (standardized effect: 0.69±0.35) and eccentric activities of inferior gluteus maximus (standardized effect: 0.23±0.35), gluteus medius (standardized effect: 0.26±0.29) and tibialis anterior (standardized effect: 0.22 ±0.06).

### Conclusions

LDT is effective in correcting overpronated feet by increasing ND height. Meanwhile, it provides a small increase in RSI and gluteal muscle activity during the eccentric (down) phase

**Funding:** The author(s) received no specific funding for this work.

**Competing interests:** The authors have declared that no competing interests exist.

of the bilateral squat, and without affecting CMJ performance. Conditioning coaches or therapists may use LDT to enhance gluteal activation for reducing injury occurrence and reactive strength performance in drop jump tasks.

## Introduction

Overpronation of the feet (flat feet) is a common foot malalignment issue where the medial longitudinal arch (MLA) is decreased, leading to increased medial plantar pressure [1]. It is reported that approximately 16.06% of Asian children aged 6 to 10 years old [2], and 13.6% of Indian adults aged 18 to 21 years old have an overpronation condition (>9 mm in navicular drop test) [3]. Research investigating the foot morphology of 196 intercollegiate athletes found that 14.4% were overpronated [4], indicating the commonality of overpronation in competitive sports. But no significant relationships were observed for athletes with overpronation issues and increased lower extremity injury prevalence [4]. Contrastingly other researchers have found associations between altered biomechanics due to lowered MLA and increased risk of chronic lower extremity injuries such as plantar fasciitis and patellofemoral pain syndrome [1, 5].

When the MLA collapses and lengthens, it directly impacts the kinetic chain of the lower limbs during weight-bearing activities (e.g., jumping and landing) [1–4]. Such issues may include tibial and femoral internal rotation, which are critical components of dynamic valgus [6, 7], which has been associated with lowered muscle activity of hip and knee stabilizers including the gluteus maximus and medius [8, 9]. Recent studies indicate a strong association between poor landing postures (hip internal rotation and valgus knee), decreased activation of knee stabilization muscles, and several common knee injuries such as anterior cruciate ligament (ACL) rupture, meniscus tear, and sprain on the medial collateral ligament in athletes [9, 10]. Therefore, overpronation may impact knee and hip kinematics during functional movements and increase the risk of knee injuries.

A common method and long-term solution to relieve undesirable stress produced by overpronation are using orthotic insoles to provide additional support to the MLA. However, tight-fitting specialized shoes (e.g., running spikes) and unfit orthotic insoles may cause additional discomfort and plantar pressure [11]. An alternative method to correct a dropped MLA is to apply low dye taping (LDT) on the plantar region, using several strips of rigid tape [12]. Research has shown that LDT significantly elevates the MLA in active asymptomatic people [13], reducing plantar pressure of the heel and medial forefoot in participants with >10 mm navicular drop (ND) [14]. Furthermore, evidence suggests that LDT can relieve pain in patients with plantar fasciitis [15]. Consequently, this evidence-informed method, in conjunction with anecdotal practice is widely accepted by practitioners in clinical settings [13, 15]. Although it is believed that knee and hip biomechanics may be improved through the correction of overpronation leading to a more efficient kinetic chain [7–10], the actual influence on proximal muscle activity, such as the gluteus maximus and gluteus medius, during functional activity is not fully understood or has been extensively studied.

Regarding the biomechanics of functional activities such as walking or running, the tightening of the plantar fascia may provide additional support to the MLA, as explained by the windlass mechanism [16]. However, the plantar fascia may be further lengthened during weight-bearing activity if the MLA collapses [17, 18]. A failed windlass mechanism in overpronated feet may affect the efficiency of absorbing ground reaction forces to produce forward propulsion during gait or accommodate uneven terrains [17]. Furthermore, it has been observed that

overpronation may cause increased mobility or hypermobility of the first and midtarsal joints compared to those with normal foot alignment and stability, thus reducing the potential to transmit force [18]. Therefore, it is postulated that correcting a dysfunctional MLA may improve foot and ankle stiffness, and subsequently athletic performance. Vertical stiffness and utilization of the fast stretch-shortening cycle can be reliably assessed using a drop jump test and calculating the reactive strength index (RSI) by dividing flight time by ground contact time [19, 20]. Whereas, the slow stretch-shortening cycle (SSC) can be determined using a countermovement jump (CMJ). Although RSI and CMJ are widely used to monitor plyometric ability for various athletes [19, 20], no study has addressed the influence of correcting the MLA of athletes with overpronation using LDT, on jumping performance.

Although there are proposed benefits of applying LDT before physical activity, further scientific support is needed for the performance change in team sport athletes such as basketball players, as prior research has predominantly focused on foot stability, joint alignment, stance, and gait [21–24]. To the author's knowledge, no previous studies measured the effects of correcting the MLA with LDT on athletic or plyometric performance. Therefore, this study aims to investigate the use of LDT to correct the MLA of recreational basketball athletes with overpronation, and investigate its effect on drop jump (including RSI) and CMJ performance, and lower extremity muscle activity during a bilateral squat.

## Methods

### Experimental approach to the problem

This study investigated how the application of LDT to correct overpronation effect on drop jump (including RSI) and CMJ performance, and lower extremity muscle activity during a bilateral squat. Participants performed all tests after non-taped (NT) and taped (TAP) conditions as a repeated crossover study design. All tests were completed in one day to maximize reliability [3]. Participants received treatment conditions (NT and TAP) in a counterbalanced order and completed tests as follows 1) drop jump, 2) CMJ and 3) bilateral squat with surface electromyography (sEMG). Muscles selected for surface EMG for the bilateral squat were superior gluteus maximus (SGMax), inferior gluteus maximus (IGMax), gluteus medius (GM), and tibialis anterior (TA). Participants were prescribed 5-minutes of rest after completing each test and another 30-minute rest was given between two conditions (NT and TAP).

**Subjects.** Twelve collegiate male recreational basketball players (age: 21.4±2.4 years [range: 19–28 years]; height: 174.5±8.2 cm; body fat: 12.4±3.6%) participated in this study. The inclusion criteria included: 1) a minimum of three years of basketball training experience (at least twice per week); 2) navicular drop (ND) ≥10 mm regarded as overpronation and; 3) body fat ≤17.5% for optimum sEMG signal. Participants with recent lower limb injury (within 12 months) or health conditions that could affect their performance or completion of this study, ND value ≤9 mm, and body fat >17.5% were not recruited. All participants completed a PAR-Q and informed consent form, and all experimental risks and benefits were disclosed. All participants passed an allergy test by putting adhesive rigid tape on the left ankle for at least 24 hours to ensure no adverse reactions. Participants were required to wear their competitive basketball shoes. This study was approved by the Human Research Ethics Committee.

**Procedures.** *Warm-up.* Before each performance test, a standardized dynamic warm-up protocol was used to increase body temperature and readiness of participants [25]. This included 30-seconds jogging, 30-seconds butt kick, 15 lunges on each leg, and 30 jumping jacks.

*Navicular drop test.* ND test procedures were adapted from Vinicombe et al. [26], which showed good intra-rater reliability with intraclass correlation coefficients (ICCs) between 0.94

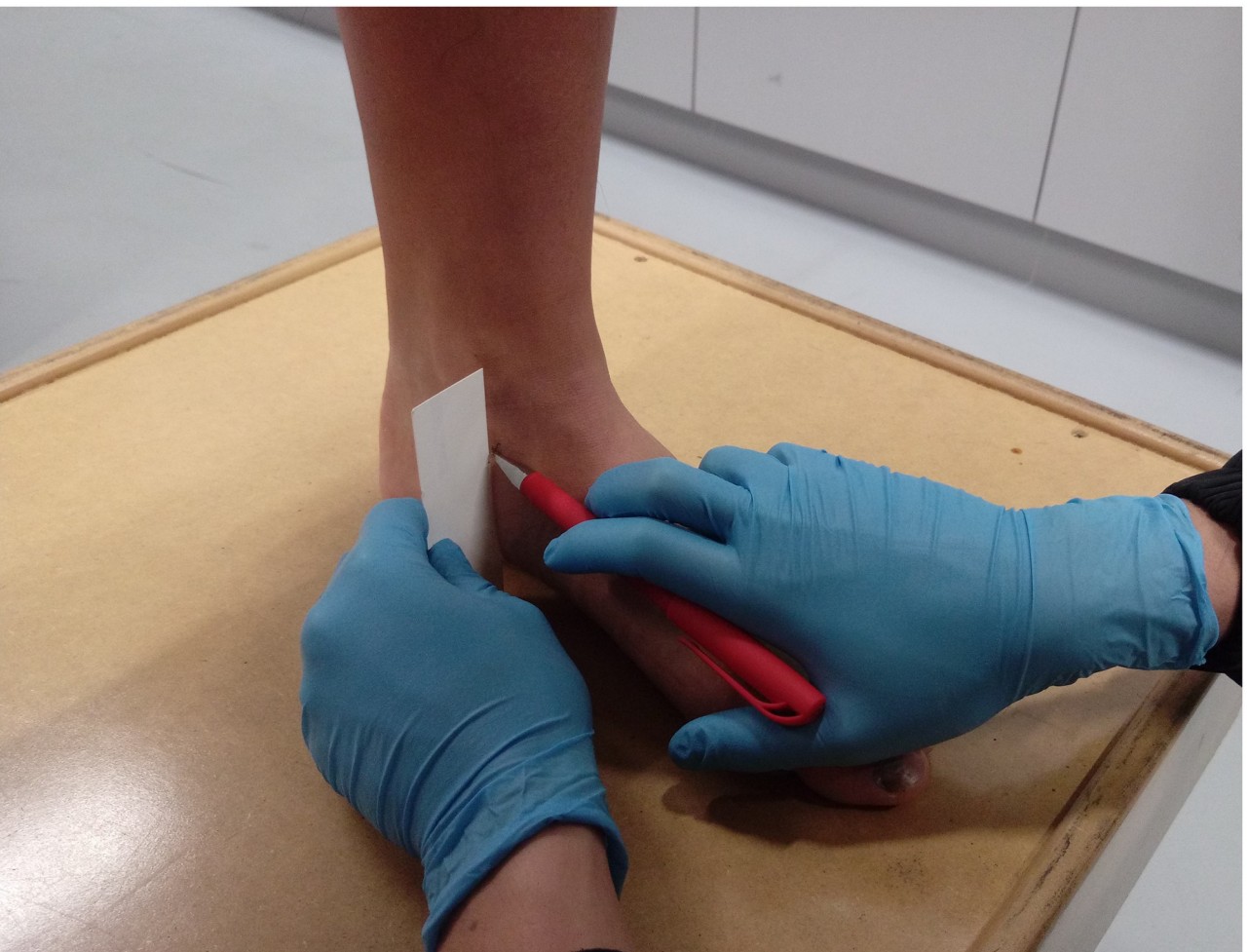

**Fig 1. Navicular drop test.**

to 0.96 [27, 28]. Participants placed their feet flat on the ground in a standing position and the navicular tuberosity was marked. To locate the ND value, participants stood naturally with weight evenly distributed on both feet, a blank card was placed perpendicular to the supporting surface against the navicular tuberosity, and the height of navicular tuberosity of the natural standing position was marked on the card. The subtalar joint neutral position was determined by putting the index finger and thumb on the medial and lateral aspects of the talus bone to palpate the talar head congruency and adjust the talar alignment until the neutral stance position was confirmed. The height of the navicular tuberosity in a neutral standing position was marked so the distance between two marks was measured as the ND value (Fig 1).

*Low-dye taping.* A standard LDT protocol was used to apply tape on participants' feet [14, 29]. Participants washed and dried their feet before starting any taping procedure. Thereafter participants placed the foot over the edge of a plinth in a supine lying position, where a strip of 3.8-cm wide zinc oxide latex-free adhesive rigid tape (Strappal, BSN Medial, Hamburg, Germany) was applied from the lateral aspect of the head of the fifth metatarsal and ran around the posterior part of the calcaneus to the medial aspect of the head of the first metatarsal as the anchor. Three to five strips (depending on foot size) of rigid tape were applied from the lateral to the medial border of the foot, with the first strip starting from the plantar region of the

forefoot proximal to the metatarsal heads. Each subsequent strip of tape overlapped the distal piece of tape by approximately half the width until the final strip was at the bottom of the ankle joint. High tension was applied when each strip was pulled to the medial border of the midfoot to support the longitudinal arch. An additional strip of tape was applied similar to the first strip to further secure all previous strips adhered (Fig 2). The administer verbally checked whether participants were comfortable after putting on shoes to confirm the completion of taping procedures.

*Drop jump test.* To assess the RSI, participants performed drop jumps (DJ) from a 40 cm platform, which is a reliable height demonstrating low typical error in RSI (coefficient of variation = 3%; ICC = 0.95) and jump height (coefficient of variation = 2.8%; ICC = 0.98) variables [30]. A stiff-legged technique was adopted due to yielding higher RSI values with decreased ground contact time [31]. A contact mat connected to Kinematic Measurement System (KMS) (Version 2014.1.2, Fitness Technology, Australia) software was used to measure RSI, which was calculated by dividing flight time by ground contact time. Participants held their pelvis with palms and index fingers touching the anterior superior iliac spine (ASIS) to minimize arm swinging (Fig 3). Two practice trials were given before testing. Participants performed three drop jump trials with maximum effort, followed by 120-second rest. To standardize drop jump performance, subjects were instructed as follows; to initiate the drop "step out" with one foot without any jumping motion, to maximize the RSI "jump as high and fast as possible while keeping knee as straight as possible" upon contact with the mat and "stay on the mat" upon second landing to avoid any errors collecting data [30]. The highest RSI value of the three trials was used for statistical analysis.

*Countermovement jump.* To measure vertical jump height, participants performed a CMJ without arm swing, which has shown ICCs between 0.87–0.99 [32]. Two retractable tape measures were taped on a smooth, flat, and vertical surface to form a 3 m measuring tape. Participants wore a headband with reflective markers placed on the bottom border to standardize all measurements. The baseline height of each participant was recorded in a natural standing position and at the point of the measuring tape adjacent to the bottom edge of the headband at the forehead region. To perform the CMJ participants were instructed to hold the pelvis with palms and index fingers touching their ASIS to minimize arm swinging, squat down to their preferred depth, then immediately jump as high as possible without any head tilt. Two practice trials were given, and participants performed 3 CMJ trials with maximum effort, followed by 120 seconds of rest [33]. A Casio Ex-100 camera (Casio Computer Co., Ltd, China) was set at the highest point achieved during practice jumps. Thereafter, slow motion video capture (240 frames per second) was used to determine the maximal jump height of each CMJ trial, which was calculated as the highest point adjacent to the reflective marker on the headband subtracted by the standing height of each subject (Fig 4).

*Bilateral squat.* Participants stood with feet shoulder-width apart and arms reaching forward with full elbow extension. Two practice trials were given, then participants performed three trials of squat with 2 seconds eccentric (down) and 2 seconds concentric (up) tempo, controlled by an electronic metronome. Thighs were required to reach parallel to the floor at the lowest squatting position, which accurate judgment was supported by applying a strip of adhesive rigid tape between the greater trochanter and lateral epicondyle of the femur (Fig 5). One minute rest was prescribed between trials.

*Surface electromyography.* During each bilateral squat muscle activity of the SGMax, IGMax, GM, and TA were measured by a 16-channel Noraxon Myosystem 1400 surface EMG unit (Noraxon Inc, Arizona, USA) at a sample rate of 1000Hz. sEMG was measured using disposable sEMG silver-silver chloride (Ag/AgCl) electrodes of 57 mm in length and 35 mm in width with conductive wet gel inside (Blue Sensor T-00-S, Ambu Inc., Denmark). For each

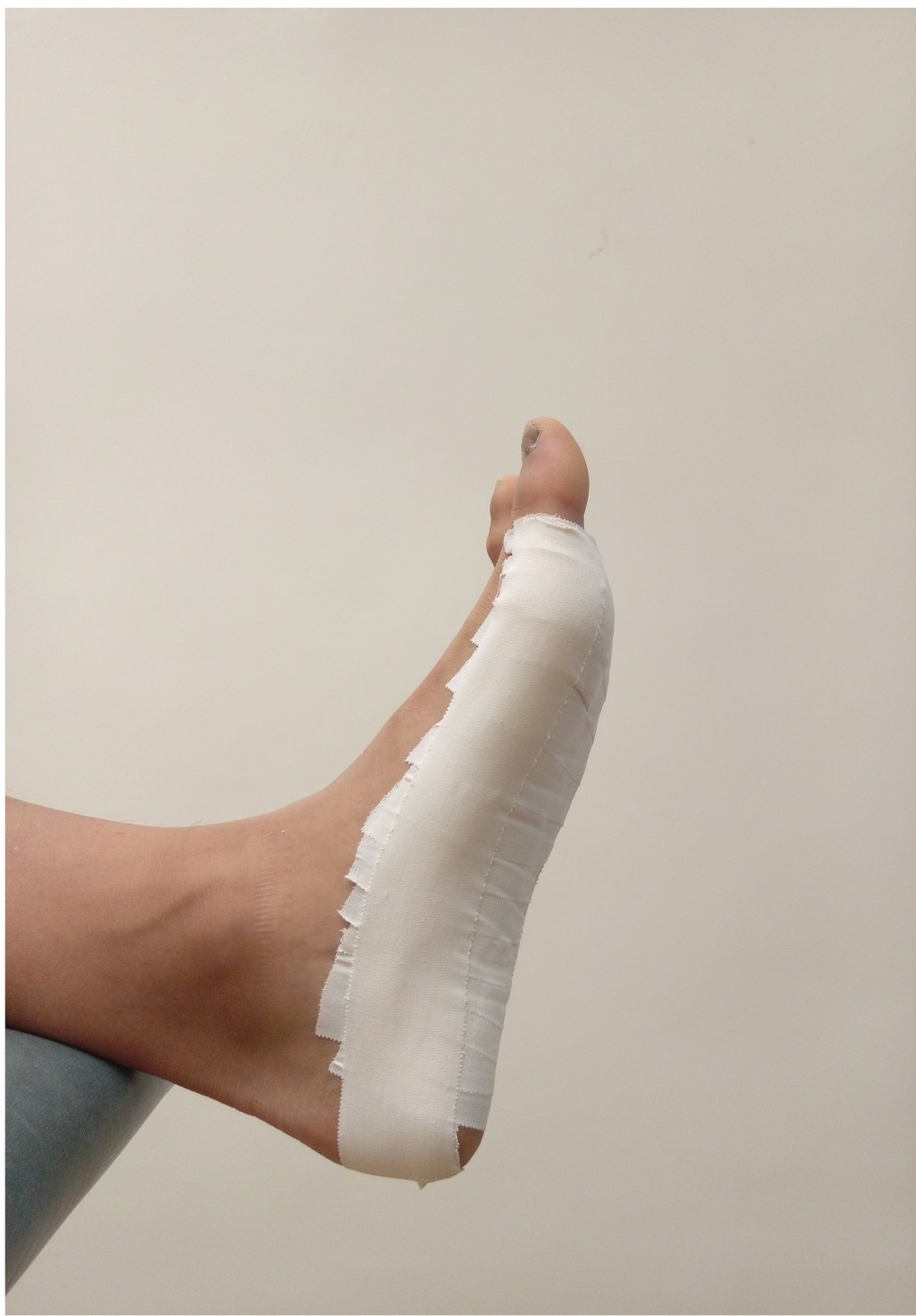

**Fig 2. The LD taping application.**

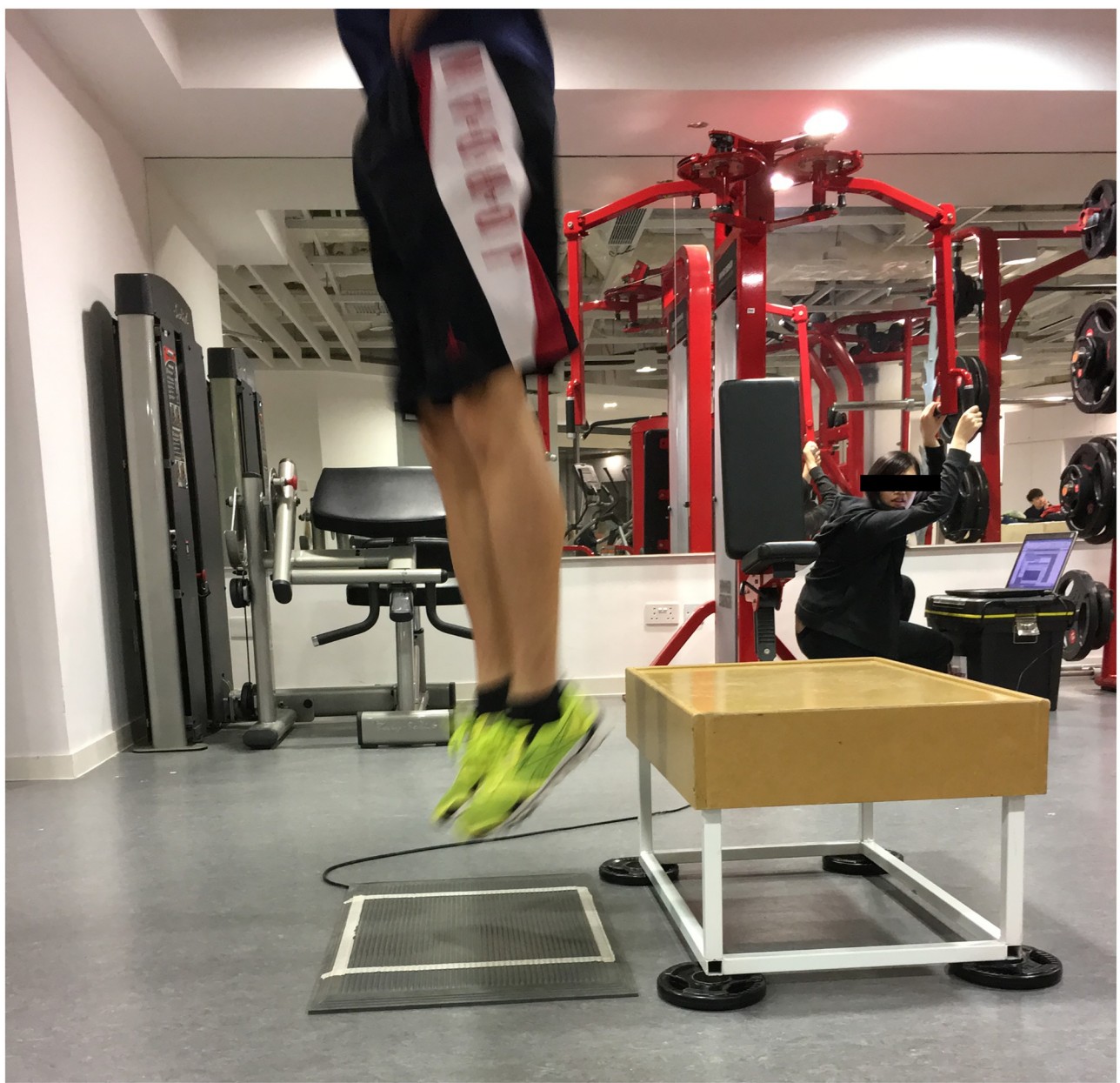

**Fig 3. Drop jump test.**

muscle, a pair of electrodes were placed with a center-to-center inter-electrode distance of 35 mm and secured with 3M™ Transpore™ surgical tape. Participants were required to shave the hair of the skin and clean it with an alcohol-soaked pad until the appearance of light redness to reduce skin impedance and optimize signal quality before electrode attachment [5].

Electrodes of the SGMax were placed superior and lateral to the line between the posterior superior iliac spine and the posterior greater trochanter, while the other two electrodes for IGMax were attached inferior and medial to the same line [34]. TA electrodes were positioned at 1/3 of the distance on the line between the tip of the fibular head and medial malleolus while GM electrodes were fixed 1/3 of the distance between the greater trochanter and iliac crest [27, 35].

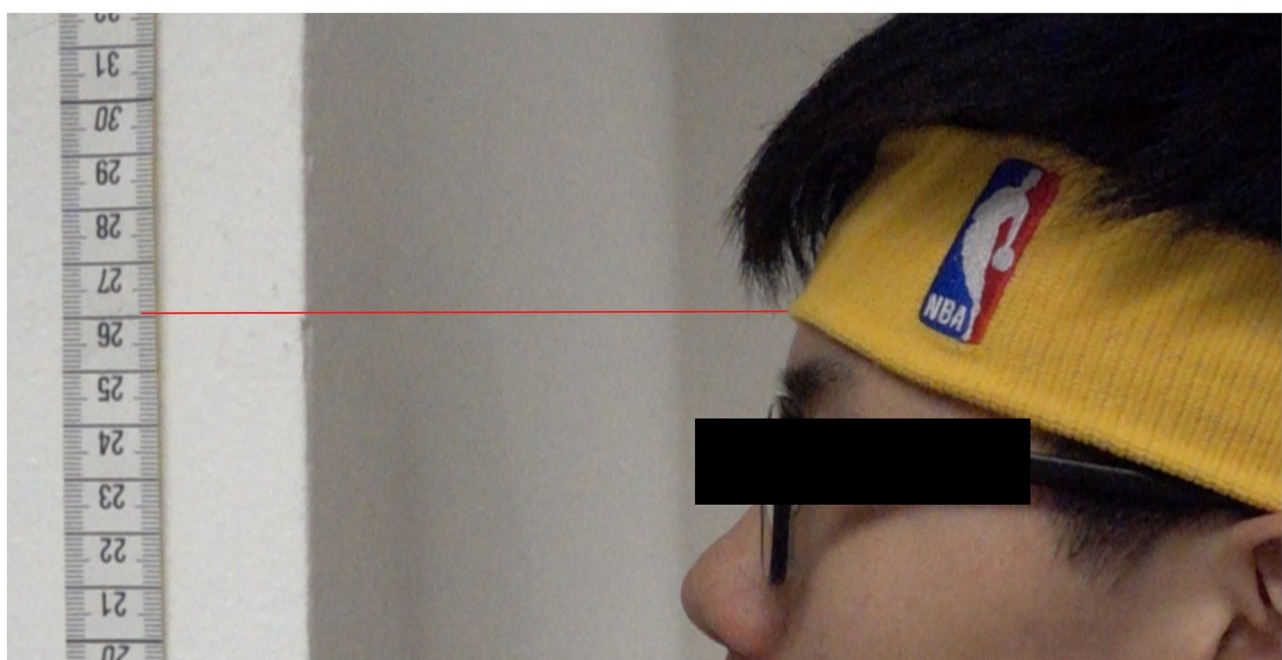

**Fig 4. Marking of non-arm swing countermovement jump.**

*sEMG normalization*. Before testing the bilateral squat, sEMG muscle activities of each muscle group being assessed were normalized using maximum voluntary isometric contraction (MVIC). The MVIC testing procedures were adopted from Kendall et al. The investigator applied manual resistance gradually until the maximum effort was attained by the participant and held for five seconds [36]. The MVIC tests of each muscle were repeated for three repetitions with one-minute rest between trials. The highest value of three repetitions was selected for further analysis. Consistent verbal encouragement was given to facilitate maximum effort throughout [35].

*Data processing for sEMG*. All raw data were processed with MyoResearch 3.8 software (Noraxon USA, Inc., Scottsdale, AZ) with full-wave rectified, band-pass filtered from 50 to 500 Hz, and smoothed via the root-mean-square (RMS) algorithm and 20-millisecond moving window. In the entire four seconds squat movement, the averaged peak activation of the initial two seconds of sEMG signals were regarded as the signals of eccentric (down) phase while the latter two seconds of sEMG signals were used to analyze the concentric (up) part of the squat.

## Statistical analysis

All sEMG data were normalized as the percentage of MVIC (%MVIC) while mean values and SDs were calculated for all variables. Shapiro-Wilk test and visual inspection were applied to assess the normality of variables. The test-retest reliability was conducted with the ICC. Paired-sample t-tests with a significance level of <0.05 were calculated for variables between NT and TAP conditions. Non-clinical magnitude-based decision (MBD) and precision of estimation were used to assess differences between taped and non-taped conditions, via respective 90% confidence intervals and the standardized effect (mean difference divided by standardized unit as Cohen's *d*). The smallest worthwhile difference was set at 0.2 and according to Hopkins et al. [37], thresholds for the magnitudes of effects were: 0.2, small; 0.6, moderate; 1.2, large; 2.0, very large; and 4.0, extremely large. The effects were unclear if the respective 90%

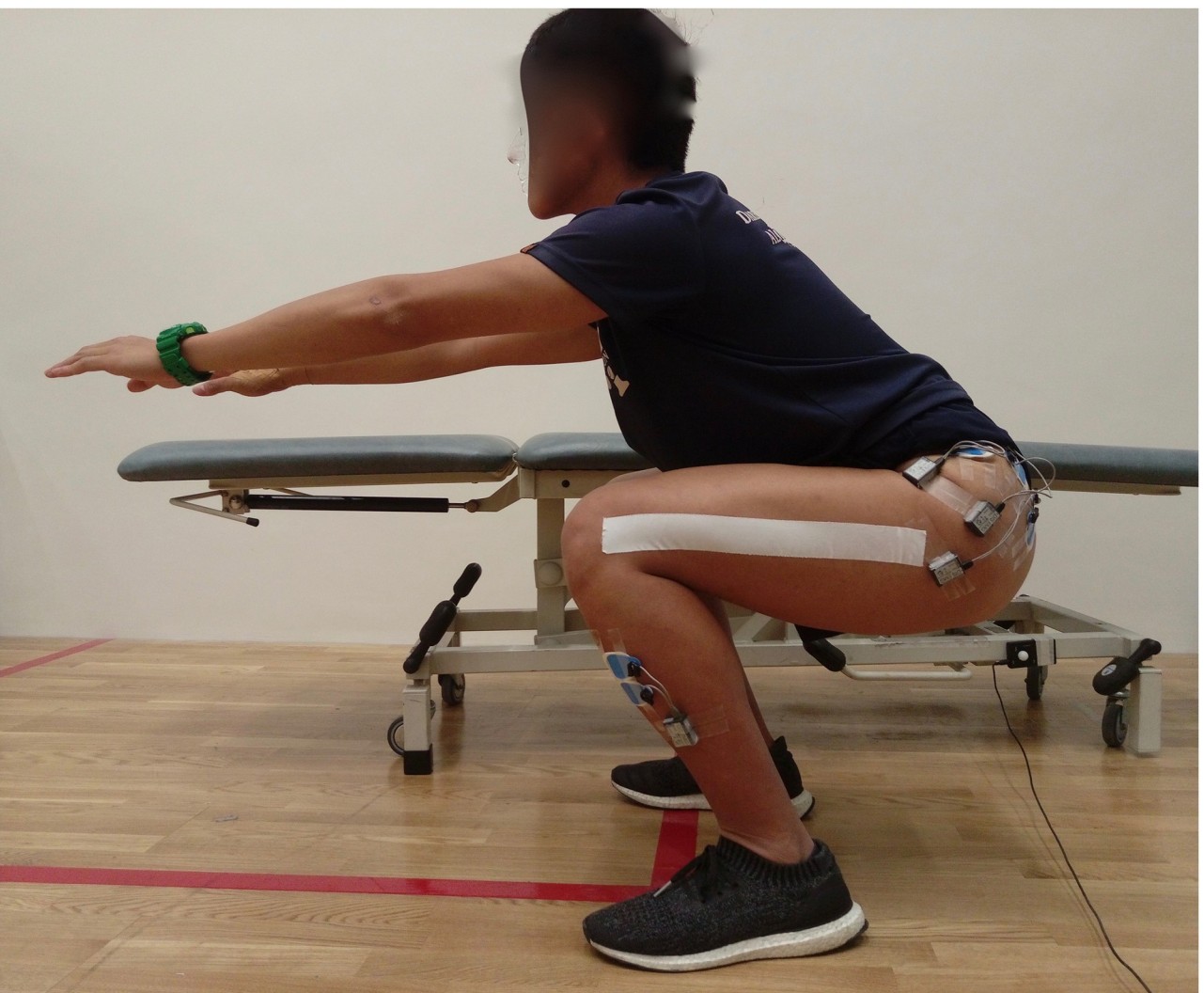

**Fig 5. Squat with electromyography testing.**

confidence intervals crossed the thresholds of the effect being substantially positive and negative by >5%. Otherwise, the effect was deemed clear with the percentage likelihood of effects being substantially negative, trivial, and substantially positive observed, and the associated qualitative inference was concluded. The probabilistic terms for classifying likelihood values were as follows: <0.5%, almost certainly not; 0.5%–5% very unlikely; 5%–25% unlikely; 25%–75% possibly; 75%–95% likely; 95%–99.5% very likely; >99.5% almost certainly [37]. All statistical analyses were conducted with RStudio software (version 1.2.5001; open-source program).

## Results

All data were normally distributed (p>0.05). All tests showed good to excellent test-retest reliability (ICC: 0.75 to 0.97) except RSI in the NT condition (ICC = 0.61), eccentric sEMG value of IGMax in NT, SGMax, IGMax, and GM in TAP conditions (ICC: 0.52 to 0.70. The paired t-test indicated that the ND value after TAP was significantly lower than that of NT condition (p<0.01) (see Table 1). Paired t-tests showed that contact time after TAP condition was

**Table 1. Comparison between non-taped (NT) and taped (TAP) conditions in performance measures.**

| Variables | NT | TAP | Difference (TAP—NT) | d | p |
|---|---|---|---|---|---|
| Navicular drop (mm) | 13.71 ± 2.73 | 9.03 ± 2.90 | -4.68 ± 0.42 | -1.54 | 0.001† |
| Countermovement jump (cm) | 45.20 ± 7.06 | 45.00 ± 6.03 | -0.13 ± 0.63 | -0.02 | 0.837 |
| Flight time of drop jump (ms) | 439 ± 39.00 | 448 ± 33.6 | 9.50 ± 6.53 | 0.24 | 0.174 |
| Contact time of drop jump (ms) | 206 ± 30.9 | 197 ± 33.5 | -9.33 ± 4.03 | -0.27 | 0.041† |
| Reactive strength index | 2.00 ± 0.24 | 2.22 ± 0.35 | 0.22 ± 0.06 | 0.69 | 0.005† |

NT = non-taped; TAP = taped;

Effect size was adjusted for small sample size.

†Significant difference between NT and TAP conditions.

significantly lower than that of NT (p = 0.041) while RSI of TAP condition was significantly higher (p<0.01). The sEMG in terms of %MVIC of all muscles between NT and TAP conditions were shown no significant difference (p>0.05) (Table 2).

The MBD results are presented in Figs 6 to 8, which shows when compared with NT, TAP provided a large effect on ND (standardized effect: -1.54±0.24), small effect on flight time (standardized effect: 0.24±0.30), small effect on contact time (standardized effect: -0.27±0.21), moderate effect on RSI (standardized effect: 0.69±0.35) and small effect on eccentric activities of IGMax (standardized effect: 0.23±0.35), GM (standardized effect: 0.26±0.29) and TA (standardized effect: 0.22±0.06).

## Discussion

Findings of this study revealed that LDT was effective in elevating the ND height for participants with overpronated feet, which was in line with the results from Holmes et al. [22]. Since the ND assesses the subtalar position [14, 22], the increase of ND height for people with dropped MLA, can therefore be considered "closer to subtalar neutral". Although the current study did not measure the change of plantar pressure, Lange et al. [14] have demonstrated the successful reduction of medial plantar pressure during walking after applying LDT. Similar findings on improving medial heel or plantar pressure during other walking tasks, drop jumps and single leg squats were also reported recently [38, 39]. It is speculated if the enhanced drop

**Table 2. Muscle activities between NT and TAP conditions (%MVIC)\*.**

| Variables | | NT | TAP | Difference (TAP—NT) | d\*\* | p |
|---|---|---|---|---|---|---|
| Eccentric | SGMax | 6.19 ± 5.21 | 6.99 ± 7.24 | 0.8 ± 0.68 | 0.12 | 0.268 |
| | IGMax | 2.59 ± 1.30 | 2.96 ± 1.73 | 0.38 ± 0.32 | 0.23 | 0.266 |
| | GM | 6.5 ± 1.83 | 7.17 ± 2.87 | 0.68 ± 0.42 | 0.26 | 0.135 |
| | TA | 33 ± 15.70 | 36.8 ± 15.9 | 3.74 ± 2.70 | 0.22 | 0.193 |
| Concentric | SGMax | 13.2 ± 5.49 | 12.8 ± 5.3 | -0.46 ± 0.80 | -0.08 | 0.577 |
| | IGMax | 10.3 ± 3.45 | 10.4 ± 3.9 | 0.1 ± 0.45 | 0.03 | 0.825 |
| | GM | 10.8 ± 4.76 | 11.3 ± 4.08 | 0.44 ± 0.67 | 0.09 | 0.528 |
| | TA | 27.4 ± 14.6 | 30.3 ± 20.1 | 2.95 ± 2.95 | 0.16 | 0.339 |

\*MVIC = maximum voluntary isometric contraction; SGMax = superior gluteus maximus; IGMax = inferior gluteus maximus; GM = gluteus medius; TA = tibialis anterior;

NT = non-taped; TAP = taped;

\*\*Effect size was adjusted for small sample size.

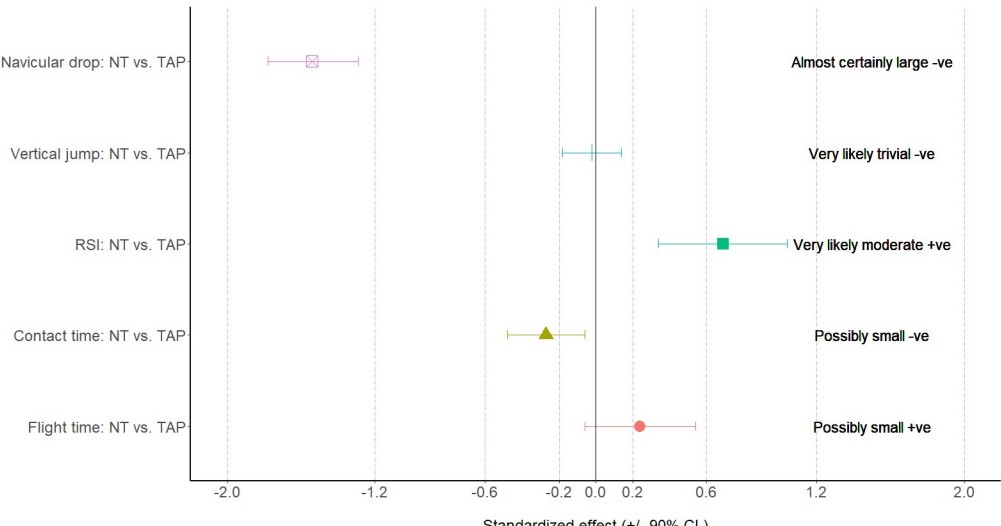

**Fig 6. Results of non-clinical magnitude-based decision in navicular drop test, countermovement jump, and drop jump performance.** RSI—reactive strength index; NT—non-taped condition; TAP—taped condition.

jump performance and muscle activity in our study are partly attributed to such plantar pressure change in our TAP condition. Further studies in this regard providing the full picture of the mechanism of how LDT potentially improves drop jump performance are warranted.

One of the important muscles to maintain subtalar stability and MLA during weight-bearing activities is the TA, which contributes to ankle dorsiflexion, deceleration of subtalar eversion, and resisting foot pronation [36, 40]. Participants with excessive foot pronation performing bilateral stability or squatting tasks without LDT may have decreased foot stability, causing higher demands on the TA to resist foot pronation [40]. Findings from Denyer et al. [41] study found no difference in TA activity during a bilateral standing task on a tilting

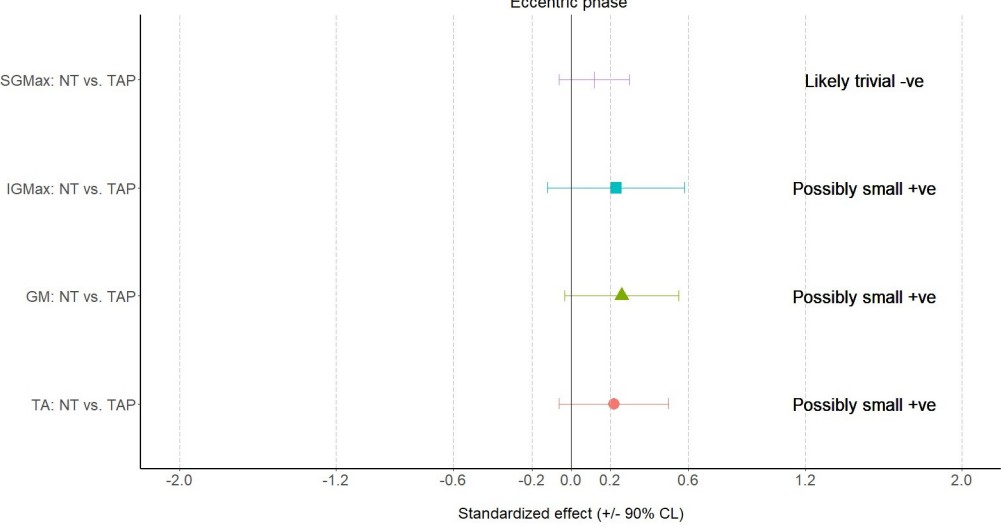

**Fig 7. Results of surface electromyography during the eccentric (down) phase of the bilateral squat.** NT—non-taped condition; TAP—taped condition; SGMax—superior gluteus maximus; IGMax—inferior gluteus maximus; GM—gluteus medius; TA—tibialis anterior.

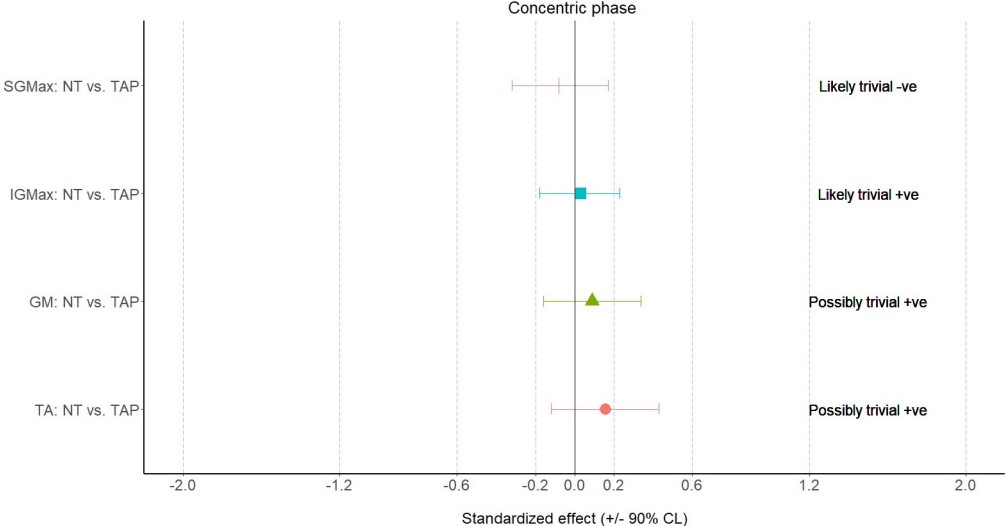

**Fig 8. Results of surface electromyography during the concentric (up) phase of the bilateral squat.** NT—non-taped condition; TAP—taped condition; SGMax—superior gluteus maximus; IGMax—inferior gluteus maximus; GM—gluteus medius; TA—tibialis anterior.

platform among individuals with excessive pronation, supination, or neutral foot structures. Despite different movement patterns, the present study demonstrated contrasting results, showing a small increase of TA activation during the eccentric (down) phase of a bilateral squat after LDT application. Performing bilateral weight-bearing tasks such as standing or squatting increased medial-lateral stability, when compared to single-leg tasks. Therefore, it is believed that additional demand on TA to maintain frontal plane stability was not needed for individuals who have a lowered MLA [42]. During the eccentric (down) phase of a squat, the TA is recruited for initiating ankle dorsiflexion and maintaining anterior-posterior stability. Excessive foot pronation without LDT support puts the TA in a lengthened position, which may decrease force production and subsequent muscle activation [43]. Similarly, in this study, small increases in IGMax and GM muscle activation were observed in the TAP condition during the eccentric (down) phase of a squat. The primary functions of IGMax and GM include hip external rotation and hip abduction, respectively. It is believed that squatting with overpronated feet could potentially increase the hip and knee internal rotation, and hence lengthen IGMax and GM muscles leading to decreased EMG signal [43], therefore, rectifying ND with LDT may promote foot supination, and tibial and hip external rotation [6, 7], and subsequently increase muscle activities by normalizing the length of the IGMax, GM, and TA.

When evaluating CMJ performance, the present findings showed no significant difference and trivial effects after using LDT. This is consistent with a recent study using foot orthoses to limit subtalar eversion and foot pronation in basketball players with or without flat feet. In their study, since only the lower limb biomechanics was altered but not the vertical jump performance [44], it was attributed to the stationary CMJ position and leading to only high reliance on the forefoot in push-off whereas other dynamic locomotions such as running and hopping have considerable midfoot and rearfoot motions. Moreover, as CMJ relies on explosive force production through triple extension (i.e., hip extension, knee extension, and ankle plantarflexion), only altering force production capabilities and kinematics in those relevant regions may affect the jumping performance [45]. However, our LDT was presumed to have no effect on the hip, knee, and ankle (talocrural joint) range of motion, meanwhile, a recent

study also proposed that the effect of only limiting inter-tarsal joints motion with LDT on vertical stiffness and hopping performance was insignificant [46]. In this study, although a small increase in IGMax, GM, and TA was observed during a slow-paced bilateral squat, such small beneficial changes induced by LDT were possibly inadequate to provide any visible kinetic or kinematic changes that would be reflected through CMJ performance.

Regarding the drop jump performance, results from this study showed a significant and moderate increase in RSI, a small decrease in contact time, and a small increase in flight time. Although stiff-legged drop jumps limit hip and knee flexion during the landing phase, they provide similar EMG muscle activation to a soft-legged technique [31]. Therefore, it is postulated that the small increase in IGMax, GM, and TA muscle activity observed in the eccentric (down) phase of the bilateral squat exercise in TAP condition, may also occur during the squatting motion in the landing phase of a drop jump. Interestingly, Struzik et al., [46] recently demonstrated no significant change in vertical ankle stiffness and hopping performance before and after LDT application on healthy basketball players. Without observing the actual change in vertical stiffness, we also assumed vertical stiffness was maintained in both ND and TAP conditions in our study. However, their study did not mention the foot alignment of the subjects and also the change of foot pronation before and after LDT application, therefore it is inconclusive regarding the relationship between frontal foot stability and hopping performance. In this regard, Porter et al., [47] suggested that faster transitions between the eccentric landing phase and concentric jumping phase (i.e., amortization phase) could reduce the loss of elastic energy, and subsequently improve jumping performance and RSI scores. A potentially shortened amortization phase and faster ground contact time observed in this study could be explained through the additional support and rigidity of the subtalar and inter-tarsal joints with LDT. However, such proposed benefits were not observed in our CMJ test and we believe that, during the slow SSC task, the vertical jump performance is mostly dependent on the hip, knee, and ankle (talocrural joint) kinematics, and the vertical stiffness while the foot frontal plane stability and the force transmission efficiency in the amortization phase on the foot region, in such a jumping task with long duration, has become insignificant and negligible.

The present study is not without limitations. Since the force production, plantar pressure, and lower extremity joint angles during jumping tasks were not measured, the actual effects of LDT on foot functions, lower limb posture and kinematics, and the force output were not fully understood. To measure the actual CMJ value, our study has adopted a novel method using a video clip to observe the true jump height, however, this method has not been validated and compared with other recognized devices such as force plate or 3D measured displacement. Moreover, only sEMG activities of the slow unloaded squatting task were observed. Future research could measure the impact of LDT on the kinematics and joint angles achieved at the hip, knee, and ankle, as well as force output and EMG activities during the vertical jump and landing tasks. Additionally, investigating the change in plantar pressure, the center of balance, and the EMG activity of foot and ankle muscles during forefoot drop landing with or without LDT would add value to this field of research. Furthermore, our study did not measure the comfort level in NT and TAP conditions and therefore, the degree of impact from such factor on performance and muscle activity were not understood.

## Practical applications

This study provides provisional evidence on the changes in lower extremity muscle activities during bilateral squat and jumping performance when using LDT on overpronated feet. Our findings offer conditioning coaches and therapists alternate techniques to potentially correct the foot alignment and enhance the fast SSC performance in drop jump without decreasing

the slow SSC performance in CMJ. Furthermore, it may give some benefits to the muscle activation of the gluteal group during squat-related activities.

## Supporting information

**S1 Data.**
(XLSX)

## Acknowledgments

The implementation of this research project was assisted by a group of year 4 sports therapy students. We especially thank Genie Tong, Kipper Lam, and Stan Chan for data collection.

## Author Contributions

**Conceptualization:** Indy M. K. Ho, Natalia C. Y. Yeung.

**Data curation:** Natalia C. Y. Yeung.

**Formal analysis:** Indy M. K. Ho.

**Investigation:** Indy M. K. Ho, Natalia C. Y. Yeung.

**Methodology:** Indy M. K. Ho, Natalia C. Y. Yeung.

**Project administration:** Natalia C. Y. Yeung.

**Software:** Indy M. K. Ho.

**Supervision:** Indy M. K. Ho, Anthony Weldon, Jim T. C. Luk.

**Validation:** Indy M. K. Ho, Anthony Weldon, Jim T. C. Luk.

**Visualization:** Indy M. K. Ho.

**Writing – original draft:** Indy M. K. Ho, Anthony Weldon, Natalia C. Y. Yeung, Jim T. C. Luk.

**Writing – review & editing:** Indy M. K. Ho, Anthony Weldon, Jim T. C. Luk.

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
