## [Decision Letter · Decision Letter 0]

19 May 2022

PONE-D-22-03490Effects of low-dye taping on drop jump and countermovement jump performance, and lower extremity muscle activity during bilateral squat in male basketball players with overpronated feetPLOS ONE

Dear Dr. Luk,

Thank you for submitting your manuscript to PLOS ONE. After careful consideration, we feel that it has merit but does not fully meet PLOS ONE’s publication criteria as it currently stands. Therefore, we invite you to submit a revised version of the manuscript that addresses the points raised during the review process.

ACADEMIC EDITOR:Dear Authors,twol experts in the field revised your manuscript retrieving several major flaws you should consider during the revision process. Please submit your revised manuscript by Jul 03 2022 11:59PM. If you will need more time than this to complete your revisions, please reply to this message or contact the journal office at plosone@plos.org. Please include the following items when submitting your revised manuscript:A rebuttal letter that responds to each point raised by the academic editor and reviewer(s). You should upload this letter as a separate file labeled 'Response to Reviewers'.A marked-up copy of your manuscript that highlights changes made to the original version. You should upload this as a separate file labeled 'Revised Manuscript with Track Changes'.An unmarked version of your revised paper without tracked changes. You should upload this as a separate file labeled 'Manuscript'.

We look forward to receiving your revised manuscript.

Kind regards,

Emiliano Cè

Academic Editor

PLOS ONE

Journal Requirements:

Reviewers' comments:

Reviewer's Responses to Questions

**Comments to the Author**

1. Is the manuscript technically sound, and do the data support the conclusions?

Reviewer #1: Yes

Reviewer #2: Partly

2. Has the statistical analysis been performed appropriately and rigorously? 

Reviewer #1: Yes

Reviewer #2: Yes

3. Have the authors made all data underlying the findings in their manuscript fully available?

Reviewer #1: Yes

Reviewer #2: Yes

4. Is the manuscript presented in an intelligible fashion and written in standard English?

Reviewer #1: Yes

Reviewer #2: Yes

5. Review Comments to the Author

Reviewer #1: The manuscript meets the publication criteria established by the journal. However, there are some issues that concern me.

TITLE

I would recommend using a more concise and impactful title. Now it's a bit repetitive and long

ABSTRACT

The objective of the study is not clearly defined in the abstract. “Effects” of DLT on what?

LDT abbreviations are not predefined in the abstract

MATERIAL AND METHODS

Was there a prior familiarization session for the tests?

Is there any information on the comfort level of individuals after taping placement? You should Indicate this like a limitation of the study, since the level of comfort could influence the performance achieved.

To determine the possible influence of fatigue, it would be interesting to report about the time that the individuals were performing the tests. 1, 2 hours or how many hours? What rest time was there between the treatment conditions (NT test and TAP)?

Line 287: superscript what does it refer to? Check and correct

RESULTS

Table 1. Indicate that the flight time refers to the drop jump, and not to the CMJ. It could confuse the reader.

DISCUSSION

General speaking, the discussion regarding the effects on performance in vertical jumps is too brief. It is necessary to provide more information about this topic. Why does DJ performance improve and not the CMJ, in which there is also a concentric phase and an eccentric phase of the movement? Could being overpronated, neutral or supinator influence the results? What has been observed in other studies? The following articles will help you in this task.

Struzik A, Stawarz M, Zawadzki J. The effect of low-Dye taping on hopping performance in handball players. Acta Bioeng Biomech. 2020;22(3):3-8. PMID: 33518734.

Radford JA, Burns J, Buchbinder R, Landorf KB, Cook C. The effect of low-Dye taping on kinematic, kinetic, and electromyographic variables: a systematic review. J Orthop Sports Phys Ther. 2006 Apr;36(4):232-41. doi: 10.2519/jospt.2006.36.4.232. PMID: 16676873

Line 390, 398 If the journal instructions do not prevent it, it is preferable to write the citation number immediately after the authors (et al.)

Line 391-393. It is speculative. Provides scientific reference and justification

Reviewer #2: Thank you for the opportunity to review this paper. I think that you should better describe the potential practical applicability of your study. Specific remarks are given below.

INTRODUCTION: consider shortening

METHODS: Subjects Exclusion and inclusion criteria must be better described (more information is needed). What were the criteria for overpronation? ND value? If so, please state that in the text and report clear cuttoff for inclusion, so that readers can understand who were the participants in this study.

Why you didn’t use contact mat for CMJ as well? And additional don’t you think that force plate would be best choice for both tests? Please elaborate that as potential limitation.

Statistics: state clearly which effect size was used

RESULTS: Table 1 and 2 are nice, clear. However, you are introducing MBI without any explanation what that is. Please correct.

DISCUSSION.

To shorten your findings you have only managed to show that LDT does change ND values in basketball players with overpronation without almost any effect on performance. In light of such findings you should go through your practical application once again to see if you data really support your claims that “Our findings offer conditioning coaches and therapists alternate techniques to reduce lower limb injuries and enhance plyometric and RSI performance. Furthermore, it provides a method for increasing muscle activation of the gluteal group during squat-related exercises without decreasing the CMJ performance.”

Also there is a study https://pubmed.ncbi.nlm.nih.gov/33518734/ that also showed no effect of LDT on mean jump height, mean ground contact time and mean vertical stiffness.

6. PLOS authors have the option to publish the peer review history of their article (what does this mean?). If published, this will include your full peer review and any attached files.

Reviewer #1: No

Reviewer #2: No

---

## [Author Response · Author response to Decision Letter 0]

31 Aug 2022

Reviewer 1

Reviewer: I would recommend using a more concise and impactful title. Now it's a bit repetitive and long

Response: Thank you for the comments and suggestions. Yes we have struggled a bit on deciding the title and were greedy to try including everything inside. Now the title is revised to more precise and appealing as “Low-dye taping may enhance physical performance and muscle activation in basketball players with overpronated feet” (line 1-2)

Reviewer: The objective of the study is not clearly defined in the abstract. “Effects” of DLT on what? LDT abbreviations are not predefined in the abstract

Response: Thank you for your suggestions. We have added (LTD) in line 40 and also indicate clearly the effects in line 42-43 as “This study investigated the effects of using low-dye taping on plyometric performance and muscle activities in recreational basketball players with overpronated feet.”

Reviewer: Was there a prior familiarization session for the tests?

Response: Thank you for your question. We did not have additional familiarization session for subjects adapting the use of low-dye taping. But we have given practice trials for the drop jump test (line 216), CMJ (line 235), squat (line 247).

Reviewer: Is there any information on the comfort level of individuals after taping placement? You should Indicate this like a limitation of the study, since the level of comfort could influence the performance achieved.

Response: Thank you for pointing out this. We didn’t measure the comfort level. Therefore we have added such limitation in line 463-465.

Reviewer: To determine the possible influence of fatigue, it would be interesting to report about the time that the individuals were performing the tests. 1, 2 hours or how many hours? What rest time was there between the treatment conditions (NT test and TAP)?

Response: Thank you for pointing this out and your reminder. We have indicated back the resting period (30 min) between conditions in line 152. Yes, we agree there can be carryover fatigue from the testing activities in the former conditions brought to the latter one. Since only a total of 5 drop jump (2 practice + 3 testing trials) and 5 CMJ (2 practice + 3 testing trials), and another 5 trials of free squat (2 practice + 3 repetitions) without failure were performed, whereas we have given resting period between tests and trials in jumping tasks and MVC tests, we assume no, or very minimum fatigue will be induced and cumulated from the first conditions to the latter one. To tackle this we performed counterbalance order for NT and TAP conditions to offset such potential bias.

Reviewer: Line 287: superscript what does it refer to? Check and correct 

Response: Thank you for pointing this out. This should be the citation but we used the inconsistent format before. Now rectified as shown in line 279

Reviewer: Table 1. Indicate that the flight time refers to the drop jump, and not to the CMJ. It could confuse the reader.

Response: Thank you for pointing this out. We have indicated “the drop jump” in table 1 for better clarity.

Reviewer: Line 390, 398 If the journal instructions do not prevent it, it is preferable to write the citation number immediately after the authors (et al.)

Response: Thank you for your reminder. We have amended per your suggestions as shown in line 382 and 394 now.

Reviewer: General speaking, the discussion regarding the effects on performance in vertical jumps is too brief. It is necessary to provide more information about this topic. Why does DJ performance improve and not the CMJ, in which there is also a concentric phase and an eccentric phase of the movement? Could being overpronated, neutral or supinator influence the results? What

has been observed in other studies? The following articles will help you in this task.

Struzik A, Stawarz M, Zawadzki J. The effect of low-Dye taping on hopping performance in handball players. Acta Bioeng Biomech. 2020;22(3):3-8. PMID: 33518734.

Radford JA, Burns J, Buchbinder R, Landorf KB, Cook C. The effect of low-Dye taping on kinematic, kinetic, and electromyographic variables: a systematic review. J Orthop Sports Phys Ther. 2006 Apr;36(4):232-41. doi:10.2519/jospt.2006.36.4.232. PMID: 16676873

Response: Thank you raising this out, your resources and suggestions. We have extended the paragraph regarding CMJ performance in line 412-427 using Struzik et al., 2020 and another paper by Ho, Kong, Chong & Lam (2019). We have also extended the content for drop jump part in line 428 to 446 while why DJ and CMJ yieided different results was explained from line 446-451.

Line 391-393. It is speculative. Provides scientific reference and justification

Response: Thank you raising this out. We have amended this part and also give additional justification with reference 38 and 39 to support such speculation (not belief). Line 383-388.

Reviewer 2

Reviewer: consider shortening

Response: Thank you for the suggestion. We have reviewed this part again and simplified/cut some sentences without affecting the meaning. The number of lines were reduced from a total of 65 lines to now 59 lines.

Reviewer: METHODS: Subjects Exclusion and inclusion criteria must be better described (more information is needed). What were the criteria for overpronation? ND value? If so, please state that in the text and report clear cuttoff for inclusion, so that readers can understand who were the participants in this study.

Response: Thank you for point out this. We have revised this section to state clearly about the exclusion and inclusion criteria including the clear cut-off point line (157-162).

Reviewer: Why you didn’t use contact mat for CMJ as well? And additional don’t you think that force plate would be best choice for both tests? Please elaborate that as potential limitation.

Response: Thank you for pointing out this. We agreed that force-plate is widely used and also kind of accepted as one of the gold-standards in jump measurement. However, recent studies shown variable jump heights using different devices. The contact mat is reliable but can underestimate high performer (Whitmer et al., 2015; PMID: 24852256). Meanwhile a very recent paper by Conceição et al., 2022 (DOI: 10.3390/app12010511) shown high jump height error/inconsistency when compared with 3D measured displacement in all devices. Our study aims to measuring the true jump height value instead of an estimated value and therefore, we use a relatively novel method. But we agree that we should add potential limitation in the discussion section and we have added this in line 455-457.

Reviewer: Statistics: state clearly which effect size was used

Response: Thank you for point out this. It is Cohen’s d and now added this back to line 299.

Reviewer: RESULTS: Table 1 and 2 are nice, clear. However, you are introducing MBI without any explanation what that is. Please correct.

Response: Thank you for your suggestion. We just found that Will Hopkins recently changed the name from magnitude-based inference to magnitude-based decision and we have also renamed all of them accordingly. Meanwhile we highlight the full form and abbreviation first in line 296 such that readers will not be confused in the result section.

Reviewer: To shorten your findings you have only managed to show that LDT does change ND values in basketball players with overpronation without almost any effect on performance. In light of such findings you should go through your practical application once again to see if you data really support your claims that “Our findings offer conditioning coaches and therapists alternate techniques to reduce lower limb injuries and enhance plyometric and RSI performance. Furthermore, it provides a method for increasing muscle activation of the gluteal group during squat-related exercises without decreasing the CMJ performance.”

Response: We have revised the practical applications to make all the claims more accurate and in line to our data/findings (Line 468-473)

Reviewer: Also there is a study https://pubmed.ncbi.nlm.nih.gov/33518734/ that also showed no effect of LDT on mean jump height, mean ground contact time and mean vertical stiffness.

Response: Thank you for giving us the recent similar study that we haven’t covered. Since in their study, they didn’t specifically recruit subjects with overpronated feet and also didn’t measure any change on the navicular alignment to check the change of medial arch, we have highlighted such differences between their and our studies in our discussion (line 434-441). 

 .

---

## [Editor Report · Decision Letter 1]

12 Sep 2022

Low-dye taping may enhance physical performance and muscle activation in basketball players with overpronated feet

PONE-D-22-03490R1

Dear Dr. Luk,

We’re pleased to inform you that your manuscript has been judged scientifically suitable for publication and will be formally accepted for publication once it meets all outstanding technical requirements.

Kind regards,

Emiliano Cè

Academic Editor

PLOS ONE
---

## [Editor Report · Acceptance letter]

2 Oct 2022

PONE-D-22-03490R1 

Low-dye taping may enhance physical performance and muscle activation in basketball players with overpronated feet 

Dear Dr. Luk:

I'm pleased to inform you that your manuscript has been deemed suitable for publication in PLOS ONE. Congratulations! Your manuscript is now with our production department. 

Kind regards, 

on behalf of

Professor Emiliano Cè 

Academic Editor

PLOS ONE